# Differences in perceptions and acceptance of COVID-19 vaccination between vaccine hesitant and non-hesitant persons

Diana Naranjo[1,2]*, Elisabeth Kimball[2], Jeanette Nelson[3], Matthew Samore[1,2], Stephen C. Alder[2,3], Kevin Stroupe[4], Charlesnika T. Evans[4,5], Frances M. Weaver[4], Cara Ray[4], Ibuola Kale[4], Patrick O. Galyean[1,2], Susan Zickmund[1,2]

1 Informatics, Decision-Enhancement and Analytic Sciences Center (IDEAS), VA Salt Lake City Health Care System, Salt Lake City, Utah, United States of America, 2 Department of Internal Medicine, Division of Epidemiology, University of Utah School of Medicine, Salt Lake City, Utah, United States of America, 3 Center for Business, Health, and Prosperity, David Eccles School of Business, University of Utah, Salt Lake City, Utah, United States of America, 4 Center of Innovation for Complex Chronic Healthcare (CINCCH) Edward Hines Health Care System, Hines, Illinois, United States of America, 5 Department of Preventive Medicine (Epidemiology), Northwestern University, Chicago, Illinois, United States of America

☽ These authors contributed equally to this work.

* diana.naranjo@utah.edu

**Data Availability Statement:** Complete interview data cannot be shared publicly because researchers assured all interview participants that their full interview responses would remain

## Abstract

Acceptance of the COVID-19 vaccination becomes more critical as new variants continue to evolve and the United States (US) attempts to move from pandemic response to management and control. COVID-19 stands out in the unique way it has polarized patients and generated sustained vaccine hesitancy over time. We sought to understand differences in perceptions and acceptance of COVID-19 vaccination between vaccine hesitant and non-hesitant patients, with the goal of informing communication and implementation strategies to increase uptake of COVID-19 vaccines in Veteran and non-Veteran communities. This qualitative study used interview data from focus groups conducted by the Department of Veterans Affairs (VA) and the University of Utah; all focus groups were conducted using the same script March-July 2021. Groups included forty-six United States Veterans receiving care at 28 VA facilities across the country and 166 non-Veterans across Utah for a total of 36 one-hour focus groups. We identified perceptions and attitudes toward COVID-19 vaccination through qualitative analysis of focus group participant remarks, grouping connections with identified themes within domains developed based on the questions asked in the focus group guide. Responses suggest participant attitudes toward the COVID-19 vaccine were shaped primarily by vaccine attitude changes over time, impacted by perceived vaccine benefits, risks, differing sources of vaccine information and political ideology. Veterans appeared more polarized, being either largely non-hesitant, or hesitant, whereas non-Veterans had a wider range of hesitancy, with more participants identifying minor doubts and concerns about receiving the vaccine, or simply being altogether unsure about receiving it. Development of COVID-19 vaccine communication strategies in Veteran and non-Veteran communities should anticipate incongruous sources of information and explicitly target community differences in perceptions of risks and benefits associated with the vaccine to

confidential, and that only data in aggregate or unidentifiable sample quotes would be included in publications and publicly shared materials related to this work. Limited data are available upon reasonable request through the University of Utah Institutional Review Board (IRB) at IRB@hsc.utah. edu or 801-581-3655.

**Funding:** This material is the result of work supported with resources and the use of facilities at the U.S. Department of Veterans Affairs (VA) Combating Antimicrobial Resistance through Rapid Implementation of Available Guidelines and Evidence (CARRIAGE) II QUERI Program and its partners at the National VA Program Offices in Hines, Illinois and Salt Lake City, Utah. It was also supported in part by the VA HSR&D Informatics, Decision-Enhancement, and Analytics Sciences (IDEAS) Center of Innovation (CIN 13-414) at VA Salt Lake City, Utah. SCA, MS, and JN received funding from the State of Utah (https://www.utah. gov/) under contract number AR3473 to collect the non-VA data analyzed in this manuscript. The funders had no role in study design, data collection and analyses, decision to publish, or preparation of the manuscript.

**Competing interests:** The authors have declared that no competing interests exist.

generate candid discussions and repair individuals' trust. We believe this could accelerate vaccine acceptance over time.

## Introduction

The Coronavirus disease (COVID-19) quickly ascended to—and remains—a leading cause of death in the United States (US), behind heart disease and cancer [1]. As COVID-19 variants continue to evolve and the US moves from pandemic response to management and control, acceptance of the COVID-19 vaccination and its related boosters remains critical. To that end, understanding perceptions of the COVID-19 vaccination during the COVID-19 Public Health Emergency (January 31, 2020- May 11, 2023) has the potential to help leaders develop more effective and persuasive COVID-19 vaccine uptake strategies, and campaigns to increase inoculation overall [2]. Though there has always been a history of variability in vaccination rates and lags in public health goals (*e.g.*, flu vaccine, pneumonia), COVID-19 stands out in the unique way it has polarized patients and generated sustained vaccine hesitancy over time [3–5]. Vaccine hesitancy is a delay in acceptance or outright refusal of a vaccine despite its availability; it is complex, can vary over time, location, and type of vaccine [6]. Indeed, vaccine hesitancy itself continues to evolve and extend its influence on other infectious diseases such as measles, which had been eliminated and considered close to eradication but is instead seeing remarkable increase in incidence [7]. COVID-19 vaccine hesitancy [8] remains a public health challenge—as disease spreads, so too, it seems, does hesitancy toward vaccination [8–10].

The availability of a COVID-19 vaccine in the US has varied and evolved over the course of the pandemic. Despite more extensive availability, widespread uptake has failed to take root. Vaccine hesitancy remains a highly documented reason for delay and/or outright refusal of the vaccine; it has become a critical building block of the landscape that constitutes our understanding of the overall pandemic and its future evolution. Other studies have documented and explored COVID-19 specific hesitancy among various populations during the pandemic era. They have found that contributors to hesitancy can vary for patients with or without physical access to vaccines. Vulnerable populations have had mixed access to COVID-19 vaccines; among those with access to a vaccine, great variability persists as to how many people ultimately get one [11]. It is also known that individuals without health care providers, in general, are less likely to be vaccinated. Preliminary evidence on sources of vaccine hesitancy also suggests that political ideologies may play a larger-than-expected role in decisions to vaccinate, independent of access. One unique aspect of the COVID-19 vaccine is the shift in political identification of those expressing hesitancy. What had been described as a sentiment among the more left-leaning spectrum of political thought is now also popular among the right—a full pendulum swing [12].

Though non-COVID-19 vaccine requirements for work and school have been in place for many years, compliance has been less robust. This is evidenced by recent increases in incidence of measles, mumps, and rubella (MMR) [7] and a resurgence of infections largely unseen since vaccination was ubiquitous. This extends outside of the US, where information about vaccinations—and hesitancy associated with them—is readily available through mass media globally, resulting in variability of vaccination adherence both for COVID-19 and other mandatory vaccinations, like MMR [13]. Current trends in non-compliance, resistance, and ongoing hesitancy towards vaccines in general foreshadow a similar struggle with the COVID-19 vaccine and infections. This presents a unique challenge for state and national vaccination

campaigns, especially in locations where vaccines abound but enthusiasm and acceptance for them does not.

We sought to assess cohort-specific differences in perceptions and acceptance of COVID-19 vaccination between vaccine hesitant and non-hesitant Utah residents and Veterans using focus groups. Our goal was that these conversations on unique barriers and insights for each cohort can guide targeted messaging meant to increase the effectiveness of vaccination campaigns for states with similar populations.

## Materials and methods

In 2021, the VA Quality Enhancement Research Initiative (QUERI) funded explorations of vaccine hesitancy to quickly assess the response to COVID-19 vaccination as part of larger effort to improve quality of care at VA hospitals across the nation. Understanding Veteran attitudes toward COVID-19 vaccine, factors associated with hesitancy, and how attitudes, beliefs, and perceptions change over time was one of the primary goals of this effort with the ultimate goal of informing communication and implementation strategies to increase uptake of vaccines among Veterans [14]. Similarly, as part of the Utah Health & Economic Recovery Outreach (HERO) project, the University of Utah conducted "Vaccine Attitude Focus Groups" to contribute additional data that would inform COVID-19 decision-making in the state. The focus here was understanding potential differences in demographically distinct populations across the state. Given that we used the same focus group interview guide, data collection, and analysis procedures, we combined data from both efforts to assess barriers and facilitators to vaccine uptake as well as differences between Veteran and non-Veteran responses. Participants across all groups were asked to discuss their attitudes toward the COVID-19 vaccine. Both Veterans and non-Veterans talked about whether they felt comfortable with the vaccine, potential risks of the vaccine, and benefits of the vaccine.

### Recruitment and sampling

We identified Veterans nationally from 28 VA facilities with Veteran vaccination rates less than 4% as of February 1, 2021, according to the VHA Support Service Center's (VSSC) "COVID Vaccine Surveillance Dashboard," which uses data from the VHA Corporate Data Warehouse (CDW), Occupational Health Recordkeeping System (OHRS), Pharmacy Benefits Management (PBM), and the "Keep Me Informed" application to track COVID-19 vaccine administration to Veterans and employees at each VA facility. These facilities were chosen because they encompassed a relatively wide geographic distribution, low vaccination rates compared to other facilities at the time, and large proportions of underrepresented Veterans (*e.g.*, African Americans, Latinx Americans, and Native Americans). Underrepresented Veterans were important to include because previous research suggests members of these groups have increased vaccine hesitancy and/or greater distrust in healthcare institutions [9,15]. Veterans with VA utilization at any of these facilities within the past 2 years, no long-term care facility stays within that period, valid mailing addresses within the 50 United States, English-speaking ability, and complete information on race and ethnicity per CDW data were included. Included Veterans received an invitation to complete a survey (which was included in the package) with questions assessing vaccination receipt/intentions. The survey also contained an option for Veteran participants to indicate interest in focus group participation. There were 10 Veteran focus groups in total, and they were organized into demographically distinct groups including women Veterans and degree of vaccine hesitancy.

Non-Veteran participants were recruited using letters and in-person contact to determine if they wished to participate in an initial survey. Survey participants were given the option to

participate in a focus group within the survey; like Veterans, those who indicated interest were contacted through email or telephone call to recruit for the focus groups. School districts, school leadership, and healthcare systems were contacted to assist in distributing recruitment letters and information for healthcare workers, educators, and high school students. Long Term Care Facilities were also contacted to assist in the recruitment of staff and flyers were posted and distributed within these facilities. In total, there were 13 demographically distinct cohorts organized with a focus on educators, medical personnel (*e.g.*, nurses, primary care physicians, long-term care facility staff), political affiliation (Republican), rural residency, Asian, Black, or African American, Pacific Islander, Hispanic, and persons who identify as women.

Focus groups were conducted remotely via Zoom platform, with 1 facilitator; the average length of the focus groups was approximately 57 minutes. Focus groups were audio-recorded, transcribed, and de-identified. Verbal consent was acquired from participants beforehand. For Veteran focus groups, the Edward Hines, Jr. VA Hospital IRB determined the present study as quality improvement. For non-Veteran focus groups, the University of Utah Internal Review Board (IRB) reviewed and determined the study to be exempt from the need for oversight.

## Focus group guide

The interview guide included questions on the effects of the COVID-19 pandemic, attitudes toward the COVID-19 vaccine, and sources of information on the COVID-19 vaccine. Women's groups were asked a question about the COVID-19 vaccine and its potential association with infertility. The interview guide is available in S1 Appendix.

## Data analysis

We used a rapid qualitative analysis process derived from the "Rapid Assessment Process (RAP) developed by J. Beebe [16] to help us develop a preliminary understanding of participant perspective and quickly develop qualitative themes. Domains were developed based on the questions asked in the focus group guide and were meant to capture themes—conceptually similar topics and ideas—discussed within the confines of those questions. Once domains were solidified, a team of two qualitatively trained analysts independently coded focus groups and met periodically to review, discuss, and adjudicate findings to establish reliability and accurate data capture. Domain summaries from all focus groups were placed into an excel spreadsheet matrix organized by domain.

Additional descriptive analyses were conducted to describe hesitancy status. Using a Likert scale, we went through each transcript and identified participants as vaccine-favorable, vaccine-unsure, and vaccine-hesitant using a scale of zero to four (0–4), with *zero* indicating full vaccine acceptance and favorability, *one* being probably vaccine favorable, *two* being vaccine-unsure, *three* indicating probably not vaccine favorable, and *four* being very vaccine-hesitant. Hesitancy status was primarily determined by 1) if the participant was vaccinated, 2) participant's response to the question regarding attitude changed toward the COVID-19 vaccine, and 3) the perceived risks and benefits of the COVID-19 vaccine. Vaccination status was also quantified for each participant, with zero being unvaccinated and one being vaccinated. Given that vaccine hesitancy [6] exists on the spectrum between high demand and full acceptance of a vaccine (i.e., 0) and complete rejection of all vaccines (*i.e.*, 4) if a participant's vaccination status was unknown (*i.e.*, they did not give a direct answer throughout the transcript and we could not determine their vaccination status with 100% certainty), their vaccination status was recorded as missing and were given at least a hesitancy score of one. They were given a score of at least one, meaning they could not be a confirmed zero (*i.e.*, no hesitancy), to account for the possibility that they were not vaccinated therefore indicating some level of hesitancy. If

participants did not respond enough for analysts to record any vaccination status, they were dropped from hesitancy scoring analyses.

## Results

A total of 36 focus groups took place with both Veterans and non-Veterans. Ten focus groups with an average of 5 participants per group (41.30% Female, 58.70% Male) took place March-July 2021 with US Veterans. More than half of Veterans in our sample were vaccinated (60.87%) and were representative of various groups: (17.39% Black/African American, 21.74% White/European American, and 19.57% Multiracial (Table 1)). Non-Veterans were engaged across 26 focus groups with participants ages 20–70, March-July 2021. 71.69% of non-Veterans represented were "White/European American" (Table 1).

Additional descriptive analyses on qualitative data allowed us to quantify vaccine hesitancy by Veteran status for 202 participants with non-missing data (44 Veterans, 158 non-Veterans) using a Likert scale. Most participants (70%) had been vaccinated at the time of the focus groups and out of those participants, a majority (53% Veterans; 73% non-Veterans) viewed the vaccine favorably (i.e., were non-hesitant). Vaccine hesitancy did not vary significantly by Veteran status (Fig 1). We also found that participants from the Black/African American community expressed favorability towards the COVID-19 vaccine, as did rural participants; vaccine hesitancy was especially pronounced in Pacific Islander participants.

Focus group responses suggest participant attitudes toward the COVID-19 vaccine were shaped primarily by time and information. The following themes emerged as salient: vaccine attitude changes over time, were impacted by perceived vaccine benefits, risks, and differing sources of vaccine information. Recurring themes are summarized in Table 2 and supported by illustrative quotations in S2 Appendix.

### Vaccine attitude changes over time

The COVID-19 pandemic has varied in its impact on vaccine hesitant individuals. Some participants had been skeptical about the vaccine when it was first announced but became more favorable towards it over time. Others felt compelled after witnessing reductions in COVID-19 deaths after vaccine distribution. Participants within the medical field often favored vaccines from the beginning and did not change their attitudes over time. This favorability occurred more often in non-Veteran groups organized with a focus on medical personnel (*e.g.*, primary care physicians, nurses, and long-term care facility staff). Others outside of the medical field were more inclined to be somewhat hesitant about receiving the vaccine at first but trended toward becoming less hesitant over time. For example, one participant shared that they were *". . .not going to receive the vaccine. And then after doing a bit of research and kind of just looking over some of the documentation on it, [they] felt like it was safe enough to go ahead. . .with it."* On the other hand, many participants (at least 10) did not exhibit a change in attitude over time; many of those who were vaccine-hesitant remained vaccine-hesitant and several (at least 5) who were vaccine-favorable remained vaccine-favorable. Others felt nervous about the vaccine and were unsure about getting it, but ultimately decided the vaccine was safer than the risk of contracting COVID-19. Non-Veteran participants reported more changes in attitudes over time though there were also non-Veteran participants that, like Veterans, did not experience changes in attitudes or beliefs over time.

### Vaccine benefits

While many participants focused on the vaccine preventing severe illness and death from the virus, some talked about how the vaccine can provide relief and benefits through other means.

**Table 1. Participant demographics by Veteran status (N = 212).**

| Non-Veteran (n = 166) | | Veteran (n = 46) | |
|---|---|---|---|
| Characteristic | N (%) | Characteristic | N (%) |
| **Gender** | | **Gender** | |
| Female | 111 (66.87) | Female | 19 (41.30) |
| Male | 50 (30.12) | Male | 27 (58.70) |
| Non-binary & non-conforming[a] | 2 (1.20) | Missing | 0 (0) |
| Missing | 3 (1.81) | | |
| **Vaccination status[b]** | | **Vaccination status[b]** | |
| | | Vaccinated | 28 (60.87) |
| | | Unvaccinated | 18 (39.13) |
| **Hispanic/Latinx** | | **Ethnicity** | |
| Yes | 19 (11.45) | Hispanic or Latinx | 9 (19.57) |
| No | 130 (78.31) | Not Hispanic or Latinx | 37 (80.43) |
| *Missing* | 17 (10.24) | Missing | 0 (0) |
| **Race** | | **Race** | |
| Black | 6 (3.61) | Black or African American | 8 (17.39) |
| White | 119 (71.69) | White or European American | 10 (21.74) |
| Multiple races | 13 (7.83) | Multiracial | 9 (19.57) |
| Pacific Islander/Native Hawaiian[a] | 11 (6.63) | Other[b] | 19 (41.30) |
| Asian[a] | 11 (6.63) | Missing | 0 (0) |
| Native American[a] | 1 (0.60) | | |
| Latinx[a] | 1 (0.60) | | |
| Missing[a] | 4 (2.41) | | |
| **Education** | | **Education** | |
| High School or Less | 12 (7.23) | High School or Less | 4 (8.70) |
| Associates Degree/Trade School | 13 (7.83) | Associate's or technical degree | 14 (30.43) |
| Bachelor's Degree | 70 (42.17) | Bachelor's degree | 11 (23.91) |
| Some college[a] | 26 (15.66) | Graduate degree[b] | 14 (30.43) |
| Master's Degree[a] | 21 (12.65) | Prefer not to answer[b] | 3 (6.52) |
| PhD/MD/Other Advanced Degree[a] | 17 (10.24) | Missing | 0 (0) |
| Unknown/missing | 7 (4.22) | | |
| **Age[a]** | | **Age[a]** | |
| 19-20s | 57 (34.34) | | |
| 30s | 39 (23.49) | | |
| 40s | 24 (14.46) | | |
| 50s | 23 (13.86) | | |
| 60s | 15 (9.04) | | |
| 70+ | 6 (3.61) | | |
| Missing | 2 (1.20) | | |

N = Total number of participants; n = number of participants in designated sample; (%) = proportion of sample

[a]This category/question was only available in the non-Veteran demographic survey.

[b]This category/question was only available in the Veteran demographic survey.

Participants across all—including Veteran vaccine-hesitant—groups mentioned less fear, more peace of mind, safety, not having to wear masks regularly, and bringing back a sense of normalcy. Expressed pros of the COVID-19 vaccine include preventing death from COVID-19, the ability to go out and gather again, and feelings of safety: "*. . .[B]enefits would just be traveling and just kind of getting back to normal as fast as possible, or as soon as possible, getting*

**VETERAN VS NON-VETERAN VACCINE HESITANCY**

**Fig 1. Veteran vs. non-Veteran vaccine hesitancy.** Distribution of vaccine hesitancy scores by Veteran status. N = 166 non-Veteran participants and N = 46 Veteran participants.

*to see your grandparents and anyone that's in immediate danger of COVID."* Vaccine-favorable participants often focused on how the vaccine would prevent one from becoming severely ill. While many did not believe it would fully protect them against the virus, they often believed it would decrease the chances of contracting COVID-19 and would prevent severe illness or death for themselves and immunocompromised members of the community.

## Vaccine risks

Overall, participants who were more vaccine-favorable viewed the risks of the vaccine as minimal and temporary and were willing to take those risks to be protected from COVID-19. Those who were vaccine-hesitant or vaccine-unsure had concerns regarding potential side effects they heard about and would not feel comfortable taking those risks with the vaccine until they felt sufficient research had been done. Vaccine-favorable participants had some concerns about the vaccine but reported ultimately preferring the risks of the vaccine to the risks of COVID-19 if they contracted the virus. For some, becoming sick after the vaccine was more of a risk because of their life and work. Several non-Veteran females across focus groups expressed concerns over potential infertility as a risk of the COVID-19 vaccine. Many vaccine-hesitant and vaccine-unsure participants believed there could be unknown side effects of the vaccine due to *"[the vaccine being] rushed too soon and there's not enough evidence."* Blood clots and myocarditis (heart inflammation) were also mentioned as specific concerns.

## Sources of information on vaccine

Overall, sources of information greatly varied across groups, though vaccine-hesitant individuals more often looked toward personal stories and distrusted large groups, such as government organizations (*e.g.*, CDC, WHO, etc.), while vaccine-favorable individuals trusted more government organizations and doctors. Both vaccine-hesitant and vaccine-favorable individuals mentioned receiving information about how and where to get the vaccine. Others discussed how they did not think there was much information on the vaccine or that the information available was often contradictory and difficult to understand. Some believed information on

**Table 2. Recurring themes and sample quotes.**

| Theme | Sample quote(s) |
|---|---|
| **Attitude changes over time** | |
| More hesitant | "I guess honestly it's just, I'm not real sure about the whole thing with my attitude over time of everyone's been pushing it so much that it makes it almost worrisome to me. . ... So it just when it almost seems like they're forcing it so much that it makes you, it makes me worried to get it. And that's the only reason that I haven't at this point because of that. I have not, no problem getting it soon in the future things like that but because it's being pushed so much it almost seems like, to me there needs to be more research on it first."<br>"I think I'm more convinced that I will not take it as time goes on." |
| Less hesitant | "I was hesitant at first mainly because of misinformation from my family. But I think I was more just willing to get it after it was being offered to us but also because I interact with a lot of people that are in the medical field. And so I was able to get some information from them too. So I think that I was more willing to get it after that."<br>"I was actually, honestly, pretty convinced I was not going to receive the vaccine. And then after doing a bit of research and kind of just looking over some of the documentation on it, I felt like it was safe enough to go ahead and go forth with it." |
| No change | "I don't necessarily think my attitude has changed. From the very beginning, when vaccines had first come out, it was kind of a little bit of an excitement that we were getting closer, and then when people started using them and nobody grew a third eye, and the zombie apocalypse didn't show up, we began to feel more and more confident in the science that these vaccines were safe and were working."<br>"For me, I've heard of some people who's had good experiences and some who've had not-so-good experiences and have had bad side effects, but for me I'm still on the nay for the vaccine. Don't want to do it just because I also feel like it's just was something that was rushed too soon and there's not enough evidence." |
| **Benefits** | |
| Feelings of safety/less fear | "Some of the benefits of the COVID vaccine is, I feel more at ease going home. . ... and there's a lot of people who are just older who have weakened immune systems. And so, for me, I feel more comfortable going home because I know I've taken appropriate measures to keep [my grandparents] as safe as possible."<br>"I can go back out again instead of being locked inside a house. . .I was just happy to be able to get out and not being worried of that if I get the COVID it's gonna kill me. That was my main concern. I can finally get outta the house." |
| Return to normalcy | "The benefits would just be traveling and just kind of getting back to normal as fast as possible, or as soon as possible, getting to see your grandparents and anyone that's in immediate danger of COVID. So it was nice and reassuring when the elderly got the vaccine and you could still hug people and just be around people. That was the good thing about it."<br>"I think a lot of people become a lot more relaxed after they got the vaccine. They tell me so. I mean it's like, I think it helps with the anxiety. . ... But I see a lot of people who just after the second dose they feel a little freer. And they definitely do feel freer. Just moving around or even getting together with relatives finally. . ... I would say that's a huge benefit for not just me but society." |
| Prevention of severe illness/death | "I would think complete immunity from being, death if you will or being hospitalized under a ventilator. I wouldn't wanna put my family nor myself through that. And the benefactor of being able to have some kind of shield of protection from a variant."<br>"I would say not 100%, especially not with the variants. So there is that little bit of a concern of getting it and maybe not knowing that I have it because I feel pretty certain that I would not become significantly ill." |
| **Risks** | |

**Table 2.** (Continued)

| Theme | Sample quote(s) |
|---|---|
| **Attitude changes over time** | |
| **Unknown risks/lack of time or data** | "I'm a pretty healthy person and I don't have any underlying conditions. And so I just feel like, if I'm doing that, and then how I got sick and the side effects or the blood clot. I'm just feeling like why risk it, to have those type of effects, when I'm not sick already, you know what I mean? And so I think that's what I'm scared of, is the long term, not knowing what will happen."<br>"For me, the biggest [risk] was—well, and this is new. It's myocarditis when your heart muscles actually swell up. That was associated with the mRNA vaccines, which is what I got. Aside from that, it's just the fear of the unknown. These are new. They were expedited. And so the long-term effects, nobody really can tell you." |
| **Mild side effects** | "Worry about the side effects or how the effects of the shot, I guess I was feeling that you know if whatever reaction I have, I'll have it. But I'm very physically strong internally so you know I think after the second injection I got maybe a 18, 20 hours of body aches and that's sometimes what I get from the flu shot. So there was no, no difference for me."<br>"I would also say, for me, it would just be the short-term risks of—I just had heard some people getting quite sick for that day. . . .. So that kind of made me a little hesitant because I just can't afford to be sick. It's just hard to be a mom and be sick. But obviously, I wanted to get the vaccine and I wanted to be vaccinated, so anyway, that was just my short-term risk." |
| **Potential long-term/serious side effects** | "Personal risks, Ball's palsy, there's been a documented list of heart issues, breathing problems, and of course there's a ton of lawsuits out there right now that the government's trying to push away for every single one of these for unknown reasons."<br>"I'm concerned about long-term risk with fertility and, even in the moment, breastfeeding and how components of the vaccine may pass through to my infant. I guess another risk is heavy metal content and heavy metal detoxification in my body post vaccine. For me, the side effects of COVID for people in my age group seem a lot better than the vaccine side effects for people, and those are the risks that I made my decision based on." |
| **Sources of Information** | |
| **Personal anecdotes** | "Well definitely don't trust the news. People, I've got some friends that are doctors and they give me some accurate information. And even they tell me not to get the vaccines. And of course with a special needs child I talk to their doctors constant, his doctors constantly. And they give me some accurate stuff about what they're seeing out there."<br>"I try to stay abreast of trends and look at some of the Journals of Medicine, but I probably put my heaviest weight on just some of the anecdotal stories, although that's not scientific, from people that I know that are hospital managers or nurses. . . .. And every time I've talked to friends that are nurses and doctors or hospital administrators, the ones that they tell me that have died, that they can talk about, from COVID, they've all had some real bad pre-existing health conditions and/or they were eventually ended up on a ventilator. And I guess these folks could have been killed if they would have gotten pneumonia or the flu." |
| **Government-sponsored individuals/organizations** | "The best information I get is from the VA. I get it through my email and they're sending updates and something on my iPad and my phone updating you on what's happening, what's available, how it's affecting people, and what they're doing to make things better. So I think the best information I get is from the VA."<br>"Well, this might not be a popular answer, but I'm just going to say Dr. Fauci. I love the guy. He's a great presence on television, and he was able to sort of keep his jobs through the whole end of the Trump era. But I don't know, you know what? I mean, he's kind of the face of the coronavirus in the United States. And he is a trustworthy guy. I like to listen to him. I like to hear what he says. And he seems to be kind of commonsense and not too radical one way or the other." |

(*Continued*)

**Table 2.** (Continued)

| Theme | Sample quote(s) |
|---|---|
| **Attitude changes over time** | |
| **Research, medical journals, doctors, etc**. . . | "I use the CDC as kind of a jumping point because I think they do a pretty decent job of simplifying it. But then, I would kind of jump into the—when I was in school, I could read about the clinical trials, I could read the academic journals and stuff, and I could understand kind of how each vaccine worked. And I never really learned which one's more effective. . .. But yeah, and just reading into kind of the offbeat cases where there's extreme side effects, I also looked a lot into the few cases where there were blood clots and stuff like that." |
| **News outlets/media** | "I was just going to say that, personally, with something that has become so politically charged, I don't feel like I can trust any media outlet at this point. There are definitely some that I feel more inclined to trust, but I just feel that because, at the end of the day, they're just trying to make money and have headlines. . ." |

the vaccine (*e.g.*, how it was made) was not readily available to them or that much of the available information was contradictory. Others, particularly those who were vaccine-favorable, felt the information they had received was readily available or enough for them to get vaccinated. Participants reported receiving information from many different sources, ranging from individuals to institutions. Sources of information include government organizations (*e.g.*, CDC, NIH, WHO), doctors, news sources, Dr. Fauci, academic journals, and people they personally know (*e.g.*, friends, family). Participants reported hearing information about the side effects of the vaccine, the availability of the vaccine, and statistics regarding vaccine effectiveness from various sources, including place of employment. While some trusted the CDC and other government entities and officials, others did not, instead relying on personal doctors and relatives for information and harboring distrust for government and media. Further, many participants felt the politicization of the COVID-19 vaccine made it difficult for people to trust information presented to them, as did the pace with which information changed over time. Several participants felt they did not receive much information on the vaccine or that the information was not transparent; social media and news outlets were also described as untrustworthy. Others felt that the information skewed positive (*i.e.*, vaccine benefits touted outweighed potential risks and side effects), fueling skepticism about the vaccine. Overall, information on the COVID-19 vaccine was scarce for some participants and it was difficult for many to find accurate and accessible information. Those who did receive information often had to seek it out or would hear about the side effects rather than other information about the vaccine. Vaccine-favorable participants believe a large pool of misinformation has contributed to vaccine hesitancy, and government officials could have done a better job addressing these problems early.

## Political ideology

In addition to the salient themes presented above, we found political ideologies also appeared to play a role in COVID-19 vaccine attitudes among all participants. Many but not all participants from the vaccine-hesitant groups expressed ideas, opinions, and ideologies often associated with right-wing ideologies, such as disliking the current President of the United States and beliefs in medical freedom. These participants also trusted religious leaders and church members (n = 3) as sources of vaccine information, an information source not mentioned by any other focus group. Similarly, many but not all participants from the vaccine-favorable groups expressed ideas and opinions often associated with left-wing ideologies, such as a

dislike for "right-wing propaganda" and a distrust of the former President of the United States.

## Hesitancy and vaccination in a nationally representative Veteran sample

The historical requirement for vaccination among military service members in the US endow Veterans with a unique lived experience, irrespective of possible personally held hesitancy with respect to vaccines. For many, their Veteran status and benefits determined how soon they could access the vaccine through the VA. In our study, Veterans appeared more polarized, being either largely non-hesitant, or hesitant. In other words, Veterans were more likely to either express no misgivings because of similar already-existing historical requirements, or to vehemently oppose vaccination because they felt this was their opportunity to have a say where they were not able to before with previous vaccines. Generally, Veterans believe in the effectiveness and safety of previous vaccines. Most Veterans did not experience marked change in attitudes and beliefs about the COVID-19 vaccine over time. Veterans relied on different sources to receive what they deemed as trustworthy information on the COVID-19 vaccine. Vaccine-hesitant Veterans often relied more on conversations with others, personal experiences, and anecdotal stories for information. These Veterans distrusted media and government organizations like the CDC to provide accurate information on the vaccine. Untrusted sources for vaccine-favorable Veterans include media (especially politically leaning or influenced), politicians, social media. Vaccine-favorable Veterans had a mix of trustworthy and untrustworthy sources of information regarding the COVID-19 vaccine. When asked about risks of the vaccine, both vaccine-hesitant and vaccine-favorable Veterans expressed side effect concerns. Overall, vaccine-hesitant, and vaccine-unsure Veteran participants were concerned with potential risks of the vaccine that may not be known. Vaccine-favorable Veterans talked about the side effects of the vaccine as risks more often, though some did not believe there were any risks inherent in receiving the vaccine. Although most Veterans did not experience a change in attitude towards the vaccine, some expressed their attitudes could change if certain conditions were met, such as more proof of its effectiveness, safety, and more documentation on potential side effects.

## Discussion

The VA has made COVID-19 vaccines—Pfizer-BioNTech and Moderna—available to Veterans, their spouses, caregivers, recipients of Civilian Health and Medical Program (CHAMPVA) benefits, and employees since Winter 2020 [17] The Utah health systems have similarly made COVID-19 vaccines accessible to patients. Veterans as servicemembers have historically been more likely to have received vaccinations and comply with mandated vaccinations, a well-known requirement for service [18]. Similarly, members of educational institutions have historically received required vaccines (*e.g.*, (MMR), polio). Despite this and the relatively high number of participants reporting vaccination in our study, enthusiastic uptake of the COVID-19 vaccine in all groups—Veterans, general patients, University staff, faculty, and students—appears to mirror that of the general US community: it is in the low and medium range with high prevalence of hesitancy and distrust [3,19]. This has and continues to make efforts aimed at mitigating the pandemic's effect on all, but especially vulnerable populations, challenging [4]

Compared to the University of Utah project with a Utah-based sample, our Veteran sample had fewer unsure individuals and proportionally more Veterans who were staunchly against the vaccine and/or would never consider getting the vaccine. Previous military experience could have influenced some beliefs. For some, it is possible that existing negative military-

specific theories, like the one suggesting multiple vaccines given to service personnel might contribute to Gulf War Syndrome [20] combined with those circulating in the community about COVID-19 [21] might be contributing to strong opposition and rejection of the vaccine among this group. Others expressed feeling more at ease with the COVID-19 vaccine because vaccines, in general, have been a long-standing service requirement [18]. Though speculative, a tension between trust and autonomy could explain the greater hesitancy of Veterans than of non-Veterans. For example, for those who had no option but to accept all vaccinations due to service requirements, refusing vaccines now presents an opportunity to exercise autonomy and control otherwise not available during service time. Similarly, transgressions by the military—a government body—occurring during a service member's tenure undoubtedly contribute to mistrust of the government and any of its efforts thereafter.

Despite ongoing public indication of vaccine safety with respect to family planning and pregnancy [22] some participants expressed concern about the vaccine causing infertility issues or interfering with family planning. Given the variability we found in patient sources of vaccine-related information, this trend was unsurprising, but an important indication of the potential ramifications misinformation may have on spurring hesitancy and reducing vaccine uptake. Government organizations and figures (*e.g.*, CDC, Fauci, etc.) were not completely trusted, even by vaccine-favorable individuals. This suggests a concerted effort by these agencies at reparation of trust might be in order, especially as we continue to navigate an uncertain future with respect to COVID-19 variants.

Population attitudes about vaccines in general have always varied, with hesitancy reigning as a critical driver in ultimate vaccine rejection among those with more negative attitudes and perceptions [6]. This remains true for the COVID-19 vaccination and at least part of what has caused continued vaccine rejection over time [3,4] The delay in acceptance or outright refusal of the COVID-19 vaccine, despite its availability, has claimed many lives in the US during the pandemic. Despite the acknowledgement of this fact by many community members, the low vaccination rates in communities are an ongoing marker that hesitancy is here to stay. Continuing COVID-19 vaccine hesitancy is likely to affect all vaccine-related health efforts, in general, thereby exerting a larger and ongoing social influence with potentially deadly consequences.

Our work has several limitations. In this work, we compared remarks from a Utah-based sample to a nationally representative Veteran sample, which, given the diverse ideological perspectives represented across individual states, might not comprehensively capture differences in perceptions of COVID-19 or vaccine-related risk compared to non-Veterans from non-Utah states. Further, we acknowledge that Black or African American participants are more represented among our Veteran respondents. We are sensitive to the social inequities and pre-existing comorbidities in this group that has led to disproportionately negative outcomes and impacts due to COVID-19 and may consequentially affect their perception of vaccine trustworthiness, especially within the context of disease control by governmental organizations and experiences of trusted peers that may be relied upon as sources of vaccine information. Lastly, we were unable to obtain vaccination status and/or presentation of comorbidity (e.g., diabetes, autoimmune disorder) as an important confounder in the choice to obtain vaccination for every participant. This made our groups not identical and therefore limited our ability to perform quantitative comparisons of the population sample. Despite the inclusion of diverse perspectives and representation from demographically distinct cohorts, most groups had largely similar themes. Given these challenges and our unique study sample, the overall generalizability and extrapolation of our findings remains limited. Despite these limitations to our work, this study offers important insight unique barriers and insights vaccine hesitant and non-hesitant persons experience. Additional information about distinct cohorts can guide targeted

messaging meant to increase the effectiveness of vaccination campaigns for states with similar populations; however, we believe this might also benefit efforts without similar populations— accumulating evidence across similar and dissimilar populations across the literature can only serve our understanding of hesitancy as a human experience and strengthen our response to it, not only regarding COVID-19, but to vaccines overall.

## Conclusions

COVID-19 vaccine communication strategies in Veteran and non-Veteran communities should anticipate incongruous sources of information and explicitly target community differences in perceptions of risks and benefits associated with the vaccine to generate candid discussions and repair patient trust. We believe this could accelerate vaccine acceptance among hesitant and non-hesitant patients over time. These findings have the potential help the development of targeted messaging than could help patients become more open to the COVID-19 and other vaccines in the future.

## Supporting information

**S1 Appendix. Focus group guide.**
(DOCX)

**S2 Appendix. Illustrative quotes.**
(DOCX)

## Acknowledgments

We would like to acknowledge and thank Howard Gordon, MD and Aaron Scherer, PhD's contributions to the focus group guide.

## Author Contributions

**Conceptualization:** Diana Naranjo, Susan Zickmund.

**Data curation:** Diana Naranjo, Elisabeth Kimball, Cara Ray.

**Formal analysis:** Diana Naranjo, Elisabeth Kimball, Cara Ray, Susan Zickmund.

**Funding acquisition:** Matthew Samore, Stephen C. Alder, Charlesnika T. Evans, Susan Zickmund.

**Investigation:** Charlesnika T. Evans.

**Methodology:** Diana Naranjo, Elisabeth Kimball, Kevin Stroupe, Frances M. Weaver, Cara Ray, Patrick O. Galyean, Susan Zickmund.

**Project administration:** Elisabeth Kimball, Jeanette Nelson, Matthew Samore, Stephen C. Alder, Charlesnika T. Evans, Cara Ray, Ibuola Kale, Susan Zickmund.

**Resources:** Stephen C. Alder, Charlesnika T. Evans, Cara Ray, Susan Zickmund.

**Supervision:** Matthew Samore, Stephen C. Alder, Susan Zickmund.

**Validation:** Jeanette Nelson, Matthew Samore, Stephen C. Alder, Frances M. Weaver, Cara Ray, Susan Zickmund.

**Visualization:** Elisabeth Kimball.

**Writing – original draft:** Diana Naranjo.

**Writing – review & editing:** Diana Naranjo, Elisabeth Kimball, Jeanette Nelson, Matthew Samore, Stephen C. Alder, Kevin Stroupe, Charlesnika T. Evans, Frances M. Weaver, Cara Ray, Ibuola Kale, Patrick O. Galyean, Susan Zickmund.

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
