## [Decision Letter · Decision Letter 0]

28 Mar 2023

PONE-D-23-05407Differences in perceptions and acceptance of COVID-19 vaccination between vaccine hesitant and non-hesitant personsPLOS ONE

Dear Dr. Naranjo,

Thank you for submitting your manuscript to PLOS ONE. After careful consideration, we feel that it has merit but does not fully meet PLOS ONE’s publication criteria as it currently stands. Therefore, we invite you to submit a revised version of the manuscript that addresses the points raised during the review process. Please submit your revised manuscript by May 12 2023 11:59PM. If you will need more time than this to complete your revisions, please reply to this message or contact the journal office at plosone@plos.org. Please include the following items when submitting your revised manuscript:A rebuttal letter that responds to each point raised by the academic editor and reviewer(s). You should upload this letter as a separate file labeled 'Response to Reviewers'.A marked-up copy of your manuscript that highlights changes made to the original version. You should upload this as a separate file labeled 'Revised Manuscript with Track Changes'.An unmarked version of your revised paper without tracked changes. You should upload this as a separate file labeled 'Manuscript'.

We look forward to receiving your revised manuscript.

Kind regards,

Andrea Cioffi

Academic Editor

PLOS ONE

Journal Requirements:

  "This research was funded by the U.S. Department of Veterans Affairs and The University of Utah. "

Additional Editor Comments:

The article has a discreet basis on which you can apply further improvements. However, it is necessary to implement various sections of the Manuscript; in consideration of the chosen topic, it will be appropriate to expand the discussions and comparison with the international literature already published on the topic.

Please note that citations recommended by reviewers may be included if you believe that they add value to your manuscript. If you do not believe that such citations would benefit your manuscript, then please provide explanation(s) in your response letter.

Reviewers' comments:

Reviewer's Responses to Questions

**Comments to the Author**

1. Is the manuscript technically sound, and do the data support the conclusions?

Reviewer #1: Partly

Reviewer #2: Partly

2. Has the statistical analysis been performed appropriately and rigorously? 

Reviewer #1: N/A

Reviewer #2: N/A

3. Have the authors made all data underlying the findings in their manuscript fully available?

Reviewer #1: No

Reviewer #2: Yes

4. Is the manuscript presented in an intelligible fashion and written in standard English?

Reviewer #1: Yes

Reviewer #2: Yes

5. Review Comments to the Author

Reviewer #1: Diana Naranjo et al. report the results of a qualitative study on COVID-19 vaccine acceptance. The study is relevant as evidence is needed for adapting risk communication strategies as COVID-19 response is transitioning from emergency to control. The paper is overall well written. However some points need clarifications/improvements.

L59-61 are about vaccine hesitancy being low both in low and high income countries, and remaining a challenge as the omicron subvariants continue to emerge. The prevalence of vaccine hesitancy has indeed been equally high both in low and high income countries as mentioned in ref(7). However this situation dates back October-November 2020. Study participants were enrolled later in 2021, and omicron variants emerged nearly a year after enrollment. By the time the omicron variant emerged, vaccine hesitancy had already evolved overtime in the US and most high income countries. It would be better to rephrase this sentence.

Throughout the results and discussion parts, veterans (from nearly country representative VAs) are compared to non-veterans from Utah. The latter may have perceived differently COVID-19 risk or vaccines related risk and benefits compared to non-veterans from other states where veteran originated. Also the distributions of ethnicity between both groups are different. E.g. Participants with black ethnicity are over represented among veterans compared to non-veterans where white participants are the most represented. Persons with black ethnicity have been heavily affected by COVID-19 given their pre-existing comorbidities and/or socio-economic status. This might positively or negatively affect their perception of COVID-19 risk resulting in either high acceptance rates or high hesitancy (e.g. mistrust in government) in a context of mistrust in official disease control bodies and important reliance on peers as trusted source of information. This weakness is not explicitly mentioned nor is its impact on the findings well addressed in the discussion.

L310-315. The authors mention several weaknesses of their study, preventing the generalizability and extrapolation of their findings. Yet in L318-320, their recommendations target all veteran and non-veterans communities. Also, the authors did not mention any study strength, making the reader question the relevancy of their findings. Could they elaborate more on this issue? Clearly showing the value added of their findings in the actual context of transition from COVID-19 emergency response to management and control.

Reviewer #2: The subject of the article - although of great interest - has already been extensively addressed in the literature in recent years. I think the study is good overall, but it requires a substantial review. In addition, it is essential that the authors discuss in a more original way the objectives and results of the study, with a greater critical spirit. This is essential for the article to be considered sufficiently innovative.

In the introduction the authors could focus more on the objectives and perspectives of their study. As rightly mentioned by the authors, vaccination hesitation is a problem that also affects other pediatric infectious diseases (MMR), whose incidence is increasing dramatically in recent years. It would be interesting and original if the authors emphasized the possible correlations between these two events.

In fact on the one hand, the problems related to the COVID-19 have reduced the availability of vaccines against other infectious diseases, especially in developing countries

(compare in this regard the article: Cioffi A, Cecannecchia C. Measles outbreaks during the COVID-19 pandemic: medico-legal and public health implications. Cad Saude Publica. 2022 Jul 25;38(7):e00095122. doi: 10.1590/0102-311XEN095122);

on the other hand, the COVID-19 - accomplice also the spread and easy availability of pandemic information through the mass media - has made adherence to the vaccine discretionary with the risk that this trend will spread to other mandatory vaccinations (or perhaps the risk is that it reinforces an existing trend?).

(compare in this regard the article: Thompson S, Meyer JC, Burnett RJ, Campbell SM. Mitigating Vaccine Hesitancy and Building Trust to Prevent Future Measles Outbreaks in England. Vaccines (Basel). 2023 Jan 28;11(2):288. doi: 10.3390/vaccines11020288).

I believe that, for example, arguing these issues (even in discussions, if necessary) can enrich the article.

I have my doubts about the methods: why did the authors not decide to submit the same questionnaire to the two groups? Some information, as noted, is only available in one of the two groups (eg vaccination status and age). This makes the two groups difficult to compare and does not allow to analyze statistically independent variables that may have affected the results.

I propose to use the same definition for the themes in Table 2 and the following paragraphs.

If Tab 2 can be an enrichment of the article, I think the arguments in the respective paragraphs are vague. For example, the authors write (line 195) "many participants did not exhibit a change in attitude over time; many of those who were vaccine-hesitant remained favorable vaccine-hesitant and several 197 who were vaccine-remained vaccine-favorable". How many? Even in the following paragraphs there are never quantitative references. Of course, I do not mean that the authors should have entered the number every time, but I think that, where there is a big difference between the two groups, the number is indispensable.

Finally, I believe that the discussions should be more expanded and argued. Although they contain interesting content, they are rather hasty. For example (but it is only one) the reasons that could explain the greater vaccination hesitation of veterans than non-veterans should be expanded.

I hope that the authors will not be discouraged by my proposed revision. On the contrary, I hope that my suggestions can be useful to make the article more appealing and original.

Finally, I thank the Editor. I am very grateful for the opportunity to review the article.

6. PLOS authors have the option to publish the peer review history of their article (what does this mean?). If published, this will include your full peer review and any attached files.

Reviewer #1: No

Reviewer #2: No

---

## [Author Response · Author response to Decision Letter 0]

10 Jul 2023

REVIEWER #1

1. Diana Naranjo et al. report the results of a qualitative study on COVID-19 vaccine acceptance. The study is relevant as evidence is needed for adapting risk communication strategies as COVID-19 response is transitioning from emergency to control. The paper is overall well written. However some points need clarifications/improvements.

Response: Thank you for these comments and for the positive feedback on the overall quality of the writing. 

2. L59-61 are about vaccine hesitancy being low both in low and high income countries, and remaining a challenge as the omicron subvariants continue to emerge. The prevalence of vaccine hesitancy has indeed been equally high both in low and high income countries as mentioned in ref(7). However this situation dates back October-November 2020. Study participants were enrolled later in 2021, and omicron variants emerged nearly a year after enrollment. By the time the omicron variant emerged, vaccine hesitancy had already evolved overtime in the US and most high income countries. It would be better to rephrase this sentence.

Response: Thank you for this comment. Our original statement implied that hesitancy was equally prevalent in both low and high-income countries; our comment on the omicron variant was meant to elevate our work’s importance; we have rephrased this sentence to better reflect the evolution of vaccine hesitancy since we collected our data and removed reference to the omicron variant. This section now reads: “Indeed, vaccine hesitancy itself continues to evolve and extend its influence on other infectious diseases such as measles, which had been considered close to eradication in the US but is instead seeing remarkable increase in incidence (7). COVID-19 vaccine hesitancy remains a public health challenge—as disease spreads, so too, it seems, does hesitancy toward vaccination (9–11).”

3. Throughout the results and discussion parts, veterans (from nearly country representative VAs) are compared to non-veterans from Utah. The latter may have perceived differently COVID-19 risk or vaccines related risk and benefits compared to non-veterans from other states where veteran originated. Also the distributions of ethnicity between both groups are different. E.g. Participants with black ethnicity are over represented among veterans compared to non-veterans where white participants are the most represented. Persons with black ethnicity have been heavily affected by COVID-19 given their pre-existing comorbidities and/or socio-economic status. This might positively or negatively affect their perception of COVID-19 risk resulting in either high acceptance rates or high hesitancy (e.g. mistrust in government) in a context of mistrust in official disease control bodies and important reliance on peers as trusted source of information. This weakness is not explicitly mentioned nor is its impact on the findings well addressed in the discussion.

Response: We have expanded our discussion to more explicitly mention the limitations in our comparisons due to the distribution of ethnicity of both groups being different and how this may impact their perceptions on COVID-19 risk or vaccines. It now reads: “Our work has several limitations. In this work, we compared remarks from a Utah-based sample to a nationally representative Veteran sample, which, given the diverse ideological perspectives represented across individual states, might not comprehensively capture differences in perceptions of COVID-19 or vaccine-related risk compared to non-Veterans from non-Utah states. Further, we acknowledge that Black or African American participants are more represented among our Veteran respondents. We are sensitive to the social inequities and pre-existing comorbidities in this group that has led to disproportionately negative outcomes and impacts due to COVID-19 and may consequentially affect their perception of vaccine trustworthiness, especially within the context of disease control by governmental organizations and experiences of trusted peers that may be relied upon as sources of vaccine information.” (pp.10-11) 

4. L310-315. The authors mention several weaknesses of their study, preventing the generalizability and extrapolation of their findings. Yet in L318-320, their recommendations target all veteran and non-veterans communities. Also, the authors did not mention any study strength, making the reader question the relevancy of their findings. Could they elaborate more on this issue? Clearly showing the value added of their findings in the actual context of transition from COVID-19 emergency response to management and control.

Response: We have added a statement describing the strengths of our findings. We also describe how, despite having a unique study sample, our findings might still be relevant to other groups outside of those identical to our own. It now reads: “Despite these limitations to our work, this study offers important insight to unique barriers and insights about vaccine hesitant and non-hesitant persons experience. Additional information about distinct cohorts can guide targeted messaging meant to increase the effectiveness of vaccination campaigns for states with similar populations; however, we believe this might also benefit efforts without similar populations—accumulating evidence across similar and dissimilar populations across the literature can only serve our understanding of hesitancy as a human experience and strengthen our response to it not only regarding COVID-19 but to vaccines overall.” (pp.11) 

REVIEWER #2

1. The subject of the article - although of great interest - has already been extensively addressed in the literature in recent years. I think the study is good overall, but it requires a substantial review. In addition, it is essential that the authors discuss in a more original way the objectives and results of the study, with a greater critical spirit. This is essential for the article to be considered sufficiently innovative. In the introduction the authors could focus more on the objectives and perspectives of their study. As rightly mentioned by the authors, vaccination hesitation is a problem that also affects other pediatric infectious diseases (MMR), whose incidence is increasing dramatically in recent years. It would be interesting and original if the authors emphasized the possible correlations between these two events. In fact on the one hand, the problems related to the COVID-19 have reduced the availability of vaccines against other infectious diseases, especially in developing countries (compare in this regard the article: Cioffi A, Cecannecchia C. Measles outbreaks during the COVID-19 pandemic: medico-legal and public health implications. Cad Saude Publica. 2022 Jul 25;38(7):e00095122. doi: 10.1590/0102-311XEN095122); on the other hand, the COVID-19 - accomplice also the spread and easy availability of pandemic information through the mass media - has made adherence to the vaccine discretionary with the risk that this trend will spread to other mandatory vaccinations (or perhaps the risk is that it reinforces an existing trend?).

(compare in this regard the article: Thompson S, Meyer JC, Burnett RJ, Campbell SM. Mitigating Vaccine Hesitancy and Building Trust to Prevent Future Measles Outbreaks in England. Vaccines (Basel). 2023 Jan 28;11(2):288. doi: 10.3390/vaccines11020288).I believe that, for example, arguing these issues (even in discussions, if necessary) can enrich the article.

Response: Thank you for bringing these two articles to our attention. We have included them in our manuscript to help illustrate the points suggested by the reviewer both in the introduction (pp.3) and in the discussion (pp.10). 

2. I have my doubts about the methods: why did the authors not decide to submit the same questionnaire to the two groups? Some information, as noted, is only available in one of the two groups (e.g., vaccination status and age). This makes the two groups difficult to compare and does not allow to analyze statistically independent variables that may have affected the results.

Response: Thank you for pointing out that this was not made sufficiently clear in our original draft. We did use the same questionnaire for both groups. In fact, it was this shared questionnaire that made the comparison of study outcomes possible. What was not identical was the types of demographic questions asked (not part of the qualitative interview guide shared by both groups). We agree with the reviewer that not having fully comparable groups in terms of information available about vaccination status and age is lamentable and does not allow for quantitative comparisons of the population sample. We have expanded our limitations section to explicitly mention this weakness in our data collection. It now reads:” Lastly, we were unable to obtain vaccination status and/or presentation of comorbidity (e.g., diabetes, autoimmune disorder) as an important confounder in the choice to obtain vaccination for every participant. This made our groups not identical and therefore limited our ability to perform quantitative comparisons of the population sample.”

3. I propose to use the same definition for the themes in Table 2 and the following paragraphs. If Tab 2 can be an enrichment of the article, I think the arguments in the respective paragraphs are vague. For example, the authors write (line 195) "many participants did not exhibit a change in attitude over time; many of those who were vaccine-hesitant remained favorable vaccine-hesitant and several 197 who were vaccine-remained vaccine-favorable". How many? Even in the following paragraphs there are never quantitative references. Of course, I do not mean that the authors should have entered the number every time, but I think that, where there is a big difference between the two groups, the number is indispensable.

Response: Thank you for this comment. We intended for the paragraphs following the table to serve as more detailed summaries of what is presented in the table. We have included numerical references where appropriate at the suggestion of the reviewer (pp.8). 

4. Finally, I believe that the discussions should be more expanded and argued. Although they contain interesting content, they are rather hasty. For example (but it is only one) the reasons that could explain the greater vaccination hesitation of veterans than non-veterans should be expanded.

Response: We have expanded the discussion to include a discussion about the vaccination hesitancy in Veterans. It now reads: “Though speculative, a tension between trust and autonomy could explain the greater hesitancy of Veterans than of non-Veterans. For example, for those who had no option but to accept all vaccinations due to service requirements, refusing vaccines now presents an opportunity to exercise autonomy and control otherwise not available during service time. Similarly, transgressions by the military—a government body—occurring during a service member’s tenure undoubtedly contribute to mistrust of the government and any of its efforts thereafter.” (pp.10). 

5. I hope that the authors will not be discouraged by my proposed revision. On the contrary, I hope that my suggestions can be useful to make the article more appealing and original. Finally, I thank the Editor. I am very grateful for the opportunity to review the article.

Response: We agree that the reviewer’s suggestions were helpful in improving the manuscript.

---

## [Decision Letter · Decision Letter 1]

11 Aug 2023

Differences in perceptions and acceptance of COVID-19 vaccination between vaccine hesitant and non-hesitant persons

PONE-D-23-05407R1

Dear Dr. Naranjo,

We’re pleased to inform you that your manuscript has been judged scientifically suitable for publication and will be formally accepted for publication once it meets all outstanding technical requirements.

Kind regards,

Andrea Cioffi

Academic Editor

PLOS ONE

Additional Editor Comments (optional):

Reviewers' comments:

Reviewer's Responses to Questions

**Comments to the Author**

1. If the authors have adequately addressed your comments raised in a previous round of review and you feel that this manuscript is now acceptable for publication, you may indicate that here to bypass the “Comments to the Author” section, enter your conflict of interest statement in the “Confidential to Editor” section, and submit your "Accept" recommendation.

Reviewer #1: All comments have been addressed

Reviewer #2: All comments have been addressed

2. Is the manuscript technically sound, and do the data support the conclusions?

Reviewer #1: Yes

Reviewer #2: Yes

3. Has the statistical analysis been performed appropriately and rigorously? 

Reviewer #1: N/A

Reviewer #2: Yes

4. Have the authors made all data underlying the findings in their manuscript fully available?

Reviewer #1: Yes

Reviewer #2: Yes

5. Is the manuscript presented in an intelligible fashion and written in standard English?

Reviewer #1: Yes

Reviewer #2: Yes

6. Review Comments to the Author

Reviewer #1: (No Response)

Reviewer #2: I appreciate that the authors implemented the article and found my suggestions inspiring. Personally I would have expanded the introduction and discussion further to allows the reader to get more into the issue; anyway, after the review, the content offers more original insights. I recognize the value of the article and further changes that I could propose risk distorting the content and style chosen by the authors. Therefore, I believe that the article so revised could be suitable for publication.

Thanks again to the Editor for the trust.

7. PLOS authors have the option to publish the peer review history of their article (what does this mean?). If published, this will include your full peer review and any attached files.

Reviewer #1: No

Reviewer #2: No

---

## [Editor Report · Acceptance letter]

29 Aug 2023

PONE-D-23-05407R1 

Differences in perceptions and acceptance of COVID-19 vaccination between vaccine hesitant and non-hesitant persons 

Dear Dr. Naranjo:

I'm pleased to inform you that your manuscript has been deemed suitable for publication in PLOS ONE. Congratulations! Your manuscript is now with our production department. 

Kind regards, 

on behalf of

Dr. Andrea Cioffi 

Academic Editor

PLOS ONE